# Essential Role of COP9 Signalosome Subunit 5 (Csn5) in Insect Pathogenicity and Asexual Development of *Beauveria bassiana*

**DOI:** 10.3390/jof7080642

**Published:** 2021-08-07

**Authors:** Ya-Ni Mou, Kang Ren, Sen-Miao Tong, Sheng-Hua Ying, Ming-Guang Feng

**Affiliations:** 1MOE Laboratory of Biosystems Homeostasis & Protection, College of Life Sciences, Zhejiang University, Hangzhou 310058, China; 11907036@zju.edu.cn (Y.-N.M.); 11807120@zju.edu.cn (K.R.); yingsh@zju.edu.cn (S.-H.Y.); 2College of Advanced Agricultural Sciences, Zhejiang A&F University, Hangzhou 311300, China; tongsm@zafu.edu.cn

**Keywords:** entomopathogenic fungi, deneddylase complex subunit, ubiquitination, gene expression and regulation, virulence, asexual development

## Abstract

Csn5 is a subunit ofthe COP9/signalosome complex in model fungi. Here, we report heavier accumulation of orthologous Csn5 in the nucleus than in the cytoplasm and its indispensability to insect pathogenicity and virulence-related cellular events of *Beauveria bassiana*. Deletion of *csn5* led to a 68% increase in intracellular ubiquitin accumulation and the dysregulation of 18 genes encoding ubiquitin-activating (E1), -conjugating (E2), and -ligating (E3) enzymes and ubiquitin-specific proteases, suggesting the role of Csn5 in balanced ubiquitination/deubiquitination. Consequently, the deletion mutant displayed abolished insect pathogenicity, marked reductions in conidial hydrophobicity and adherence to the insect cuticle, the abolished secretion of cuticle penetration-required enzymes, blocked haemocoel colonisation, and reduced conidiation capacity despite unaffected biomass accumulation. These phenotypes correlated well with sharply repressed or abolished expressions of key hydrophobin genes required for hydrophobin biosynthesis/assembly and of developmental activator genes essential for aerial conidiation and submerged blastospore production. In the mutant, increased sensitivities to heat shock and oxidative stress also correlated with reduced expression levels of several heat-responsive genes and decreased activities of antioxidant enzymes. Altogether, Csn5-reliant ubiquitination/deubiquitination balance coordinates the expression of those crucial genes and the quality control of functionally important enzymes, which are collectively essential for fungal pathogenicity, virulence-related cellular events, and asexual development.

## 1. Introduction

Hypocrealean insect pathogens are main sources of fungal insecticides, which are environment friendly and safe to apiculture [1,2]. The biological control potential of such fungi involves the overall output of various cellular processes that are determinant to the virulence, stress tolerance, and conidiation capacity vital for survival/dispersal in host habitats and are controlled or affected by numerous effectors and signalling proteins [3,4,5,6]. The ubiquitination of target proteins is one of the posttranslational modification (PTM) mechanisms underlying the overall output [7,8] and is regulated by the ubiquitin–proteasome system (UPS), which comprises ubiquitin-activating (E1), -conjugating (E2) and -ligating (E3) enzymes and the ubiquitin-specific proteases (USPs) or deubiquitinases required for protein quality control and cell growth, division, differentiation, and development [9,10,11].

The COP9 (constitutive morphogenesis number 9 [12]) signalosome (CSN) is an eight-ubunit (Csn1–8/CsnA–H) protein complex that acts as a core player in PTMs such as ubiquitination/deubiquitination [13] and functions in diverse pathways essential for transcriptional regulation, DNA repair, the cell cycle, and cell differentiation and development across eukaryotes [14,15,16,17,18]. Cullins serve as the substrates of CSN and the subunits of cullin ring E3 ligases (CRLs), and as a scaffold for the formation of multi- subunit complexes [19]. The activation of CRLs relies upon the neddylation of cullins by Nedd8/NeddH, a ubiquitin-like protein [20,21]. CRL activity is controlled through attachment/removal cycles of Nedd8, which act as a switch for CRL-dependent ubiquitination for the activation or repression of target function [22].

CSN subunits have been characterised in model fungi. Subunit-deficient mutants of fission yeast showed distinct phenotypes, implicating the functional differentiation of CSN subunits [23,24]. In budding yeast, single-gene knockout mutants of four proteins interacting with Csn5 led to increased cullin accumulation such as blocked deneddylation in fission yeast ∆*csn* mutants [25], suggesting a conserved role of CSN in mediating cullin deneddylation. Defective CSN assembly or activity resulted in decreased quantities of ergosterol and unsaturated fatty acids, vacuole defects, diminished sizes of lipid droplets, and accumulated endoplasmic reticulum stress [26]. In *Candida albicans*, the protein CAND1 (cullin-associated, neddylation-dissociated) was shown to trigger SCF (Skp1–cullin/ Cdc53 –F-boxprotein) ubiquitin ligase activity in a fashion independent of neddylation [27]. In *Neurospora crassa*, the integrity of CSN was crucial for hyphal growth, asexualdevelopment, and circadian function because site mutations of the JAMM (JAB1/MPN/ Mov34 metalloenzyme) domain required for deneddylase activity [28] disrupted its deneddylation activity despite little impact on the CSN assembly or CSN–cullin interactions and mitigated phenotypic defects in ∆*csn5*, in which clock protein FRQ was partially degraded, while the substrate receptors of CRLs were not degraded [29]. In *Aspergillus nidulans*, single-gene deletion mutants of CSN subunits displayed identical phenotypes including the production of aberrant red pigment and the incapability to form mature sexual fruit bodies [30,31], although severe growth defects of ∆*csnD* and ∆*csnE* occurred under DNA-damaging stress [32]. NeddH recruits all three cullins and is removed from cullins by CSN [33]. Inactivation of CsnE containing the conserved JAMM domain caused more accumulation of neddylated proteins [30]. The site-mutated core of the JAMM domain resulted in the same phenotype as the ∆*csn* mutants, although CsnE failed to recruit another subunit in the absence of *csnA* or *csnD* [31,32]. Omics analyses revealed that CSN activity affected the transcription of oxidoreductases, the production of developmental hormones and secondary metabolites, and the rearrangement of the cell wall in *A. nidulans* [34]. The light-responsive development of *A. nidulans* is evidently controlled by the interacting deneddylases Den1/DenA and CSN because their interaction affected Den1/DenA levels and deneddylase activity [35]. Indeed, each CSN subunit has proved essential for fungal deneddylase activity, which was lacking in each of the *A. nidulans* ∆*csn* mutants compromised similarly regarding their development and secondary metabolisms [36]. The latter study revealed the formation of a seven subunit, pre-CSN intermediate without catalytic activity and the restoration of deneddylase activity through the integration of CsnE into the intermediate, indicating that the integration was a final step for activating CSN. Recently, the ubiquitin-specific protease UspA, homologous to human CSN-associated Usp15, was found to interact with six CSN subunits, reduce amounts of ubiquitinated proteins during development, and accelerate conidiation and sexual development in *A. nidulans* [37,38], implying an interplay between CSN deneddylase and UspA deubiquitinase. These studies indicate conserved activities of CSN subunits and, more or less, differentiation of their functions in model fungi and suggest their links to ubiquitination/deubiquitination.

Aside from intensive studies in model fungi, Csn5 was not explored in filamentous fungal pathogens until recently. As very limited examples, CsnE proved essential for conidiation and secondary metabolism in *Pestalotiopsis fici* because deletion of *csnE* led to abolished conidiation, the facilitated and reduced production of chloroisosulochrin and ficiolide A, respectively, and the dysregulation of 8.37% of the genes in the whole fungal genome [39]. Similarly, the deletion of *csn5* in *Alternaria alternata* resulted in a moderate growth defect, abolished conidiation, lost plant pathogenicity, and dysregulation of 1658 genes during conidiation and of 1787 genes during infection [40]. The limited studies unveil the significance of Csn5/CsnE for the fungal lifecycle in vitro and in vivo, suggesting thenecessity of exploringthe conserved and special roles of its orthologues in many other pathogenic fungi, which have adapted to various hosts and habitats along evolutionarily distinctive trajectories. To date, no effort has been made to characterize Csn5 in fungal insect pathogens that could have evolved insect pathogenicity from plant pathogens or endophytes ~200 million years ago [1]. This study sought to elucidate the roles of orthologous Csn5 in *Beauveria bassiana*, a classic insect pathogen used as a main source of wide-spectrum fungal insecticides. As presented below, Csn5 plays a prominent role in the balance of ubiquitinationand and deubiquitination through the transcriptional coordination of multiple UPS genes. This Csn5-reliant balance coordinates the functions of the crucial genes and proteins or enzymes required for cell hydrophobicity, asexual development, and insect pathogenicity as well as the response to hydrogen peroxide and heat shock in *B. bassiana*.

## 2. Materials and Methods

### 2.1. Recognition and Bioinformatic Analysis of Fungal Csn5 Orthologues

The amino acid sequence of *A. nidulans* CsnE (Q5BBF1) was used as a query to search through the NCBI databases of *B. bassiana* and other entomopathogenic and nonentomopathogenic fungi using BLAST analysis (https://www.ncbi.nlm.nih.gov/ (accessed on 5 August 2021)). The identified orthologues were subjected to conserved domain analysis online onthe web site https://www.ncbi.nlm.nih.gov/Structure/ (accessed on 5 August 2021) and phylogenetic analysis with the MEGA7 program at http://www.megasoftware.net/ (accessed on 5 August 2021), followed by predicting the nuclear localisation signal (NLS) motif from each protein sequence at http://nls-mapper.iab.keio.ac.jp/ (accessed on 5 August 2021).

### 2.2. Subcellular Localisation of Csn5 in B. bassiana

Transgenic strains expressing the fusion protein Csn5-GFP in wild-type strain *B. bassiana* ARSEF 2860 (designated WT) were constructed using backbone plasmid pAN52-C-gfp-bar, where C denoted cassette 5′-*Pme*I-*Spe*I-*Eco*RV-*Eco*RI-*Bam*HI-3′, which was controlled by the homologous promoter P*tef1* [41,42]. Briefly, the coding sequence of Csn5 was amplified from the WT cDNA with paired primers (Appendix A) and inserted into the N-terminus of *gfp* (green fluorescence protein gene) in the linearised plasmid. The new plasmid, pAN52-csn5-gfp-bar, was integrated into the WT via *Agrobacterium* mediated transformation. Putative transformants were screened by *b**ar* resistance to phosphinothricin (200 μg/mL). A transformant expressing the desired green signal was chosen for incubation on SDAY (Sabouraud dextrose agar (4% glucose, 1% peptone, and 1.5% agar) plus 1% yeast extract) for conidiation. The conidia were incubated in SDBY (i.e., agar-free SDAY) on a shaking bed (150 rpm) at 25 °C for 48 h. Culture samples were stained with the nuclear dye DAPI (4′,6′-diamidine-2′-phenylindole dihydrochloride; Sigma) and visualised for the subcellular localisation of Csn5-GFP using laser-scanning confocal microscopy (LSCM). The green fluorescence intensities in the cytoplasm and nucleus of each cell in nine hyphae were assessed with ImageJ (https://imagej.nih.gov/ij/ (accessed on 5 August 2021)). The nuclear versus cytoplasmic green fluorescence intensity (N/C–GFI) ratios were computed as the relative accumulation levels of Csn5-GFP in the nuclei of examined hyphae.

### 2.3. Generation of csn5 Mutants

For targeted gene disruption, the 3′ and 5′ coding/flanking fragments (1371 and 1154 bp, respectively) of *c**sn5* were amplified from the WT DNA and inserted into the *Xma*I/ *Bam*HI and *Xba*I/*Hpa*I sites ofp0380-bar. The new vector, p038-5′csn5-bar-3′csn5, was transformed into the WT strain, as aforementioned, for the disruption of *c**sn5* through the deletion of a promoter/coding fragment (43/429 bp) via the homologous recombination of the *b**ar*-separated fragments. Subsequently, the full-length coding sequence of *C**sn5* with flanking regions (3991 bp in total) was amplified from the WT DNA and ligated to the sites of *Hin*dIII/*Xba*I in the p0380-sur-gateway in exchange for the gateway fragment. The resultant p0380-sur-csn5 was ectopically integrated into an identified Δ*csn5* mutant for targeted gene complementation. Putative mutants were screened by their *bar* resistance to phosphinothricin (200 μg/mL) or their *sur* resistance to chlorimuron ethyl (10 μg/mL). Expected recombinant events in the colonies were identified via PCR and real-time quantitative PCR (qPCR). Listed in Appendix A are the paired primers used for targeted gene manipulation. The positive mutants, ∆*csn5* and ∆*csn5::**C**sn5* (Appendix A), were evaluated together with the parental WT in the following experiments of three independent replicates, which generated data meeting the requirements for one-factor analysis of variance and Tukey’s honestly significant difference (HSD) test between the mutants and WT.

### 2.4. Western Blot

Intracellular free ubiquitin accumulation levels were assessed from the protein extracts isolated from the 3-day-old SDBY cultures, as described previously [7]. Briefly, aliquots of 20 μg protein extracts were loaded onto 12% SDS-PAGE and transferred to polyvinylidene difluoride (PVDF) membranes, followed by western blotting with 2000-fold dilution of rabbit monoclonal anti-ubiquitin antibody (Boster, Wuhan, China; catalogue number BM4359). The internal standard was based on the blots of β-actin probed with 5000-fold dilution of anti-β-actin mouse monoclonal antibodies (FudeBiological Technology, Hangzhou, China; catalogue number FD0060). The bound antibodies were reacted with 5000-fold dilution of horse radish peroxidase (HRP)-conjugated anti-rabbit antibodies (Boster; catalogue number BA1054) and visualised in a chemiluminescence detection system (Amersham Biosciences, Shanghai, China). A blot gel chosen from three western replicates was presented. The ratios of ubiquitin versus β-actin blot intensities assessed with ImageJ were computed as relative ubiquitin accumulation levels in the extracts of each strain.

### 2.5. Bioassays for Fungal Virulence

The virulence of each strain via normal cuticle infection (NCI) or cuticle-bypassing infection (CBI) was assayed by immersing three groups of ~35 third-instar *Galleria mellonella* larvae for 10 s in 40 mL aliquots of a 10^7^ conidia/mL suspension or injecting 5 μL aliquots of a 10^5^ conidia/mL suspension into the haemocoel of each larva in each group. Three groups of larvae immersed in or injected with 0.02% Tween-containing sterile water (used in conidial suspension) were used as controls. All treated groups were maintained at the optimal temperature of 25 °C and monitored every 12 h for survival/ mortality records. The time–mortality records in each group were corrected using the background mortality in the controls, followed by probit analysis for estimation of the median lethal time (LT_50_) as a virulence index.

### 2.6. Examination of Cellular Events Associated with NCI and Virulence

Conidial adherence to the insect cuticle was assessed using pre-treated locust hind wings, as described elsewhere [43]. Briefly, suspensions of 10^7^ conidia/mL surfactant-free sterile water were prepared by thorough vortex, and 5 μL aliquots of each were spotted onto the central areas of the hind wings attached to 0.7% water agar. After an 8 h incubation at 25 °C, the counts of the conidia in the three microscopic fields of each wing were made before and after less-adhesive conidia were washed for 30 s in sterile water. Conidial adherence to the wing cuticle was computed as percent ratios of pre-wash versus post-wash counts with respect to the WT standard.

The larvae that were dead post-CBI were maintained at 25 °C for hyphal outgrowth as an indicator of the hyphal capability for penetrating through the insect cuticle. Total activities of the extracellular enzymes (ECEs, including those involved in proteolysis, chitinolysis, and lipolysis) and the subtilisin-like Pr1 family proteases collectively required for NCI [44,45] were quantified from the supernatants of 3-day-old cultures initiated with a 10^6^ conidia/mL suspension in CDB (Czapek-Dox broth: 3% sucrose, 0.3% NaNO_3_, 0.1% K_2_HPO_4_, 0.05% KCl, 0.05% MgSO_4_, and 0.001% FeSO_4_) and amended with the sole nitrogen source of 0.3% bovine serum albumin (BSA) as an enzyme inducer, as described previously [45,46]. The presence and abundance of hyphal bodies in the haemolymph samples taken from surviving larvae post-NCI or post-CBI were examined with microscopy. Biomass levels and blastospore concentrations were measured from the 3-day-old cultures initiated with a 10^6^ conidia/mL suspension in insect haemolymph- mimicking trehalose-peptone broth (TPB; i.e., CDB amended with 3% trehalose as the sole carbon source and 0.5% peptone as the sole nitrogen source) to assess the dimorphic transition rate crucial for the acceleration of intrahaemocoel cell proliferation and host death through yeast-like budding. Flow cytometry was performed to reveal carbohydrate epitope patterns on the surfaces of 5 × 10^4^ conidia or blastospores (in each of three samples per strain) labelled with the Alexa Fluor 488-labelled lectins ConA (concanavalin A), WGA (wheat germ agglutinin), PNA (peanut agglutinin), and GNL (*Galanthus nivalis* lectin) (Vector Laboratories, Burlingame, CA, USA), respectively, or the cell cycles of blastospores stained with propidium iodide (DNA dye), as described elsewhere [47].

### 2.7. Assays for Radial Growth, Stress Response, Conidiation Capacity, and Conidial Quality

To revealtheimpact of *csn5* disruption on hyphal growth and invasion into the insect body, 1 μL aliquots of a 10^6^ conidia/mL suspension were spotted onto SDAY and CDA (i.e., CDB plus 1.5% agar) plates, followed by an 8-day incubation underthe optimal regime of 25 °C in a light/dark (L:D) cycle of 12:12 h. Cellular responses to stress cues likely encountered duringNCI and haemocoel colonisation were assayed using 8-day-old colonies initiated as above on CDA alone (control) or supplemented with NaCl (0.4 M) or sorbitol (1.5 M) for osmotic stress, menadione (0.02 mM) or H_2_O_2_ (2 mM) for oxidative stress, and Congo red (6μg/mL) or calcofluor white (10 μg/mL) for cell wall stress, respectively. Additionally, 2-day-old SDAY colonies initiated at 25 °C were exposed to 42 °C for a 3 h heat shock, followed by a 6-day growth recovery at 25 °C. The diameter of each colony was estimatedwith two measurements taken perpendicular to each other across the centre. Cell sensitivity to each stress was calculated as the percentage of relative growth inhibition (RGI = (*d*_c_ − *d*_s_)/*d*_c_ × 100) using control (*d*_c_) and stressed (*d*_s_) colony diameters.

To assess conidiation capacity and biomass accumulation, 100 μL aliquots of a 10^7^ conidia/mL suspension were spread on SDAY plates (9 cm diameter) overlaid with or without cellophane and incubated for 9 days under the optimal regime. The conidiation status of each strain was microscopically examined using the samples taken from 3- or 6-day-old cultures and stained with calcofluor white (cell wall-specific dye). From day 5 onward, conidial yield in three samples taken every 2 days from each plate culture with a cork borer (5 mm diameter) was quantified as the number of conidia per unit area (cm^2^), as described previously [48,49]. Meanwhile, biomass levels were measured from cellophane-overlaid SDAY cultures every 2 days from day 4 onward. Hydrophobic/hydro- philic features of such cultures were observed by dropping 10 μL aliquots of 0.02% Tween 80 onto the surfaces of 4-day-old SDAY cultures and monitoring the dispersal of water droplets for 72 h at 25 °C, as described previously [50]. In addition, the total activities of superoxide dismutases (SODs) and catalases (U/mg) were assayedfrom the protein extracts of 3-day-old SDAY cultures with an SOD Activity Assay Kit (Sigma, Shanghai, China) and a Catalase Activity Assays Kit (Jiancheng Biotech, Nanjing, China) following the manufacturers’ guides, respectively.

The quality of conidia from the 9-day-old SDAY cultures was assessed using the indices of median germination time (GT_50_, h) at 25 °C, the median lethal dose (LD_50_, J/cm^2^) for UVB (weighted wavelength: 312 nm) resistance and hydrophobicity in an aqueous/ organic system, and examined with scanning electronic microscopy (SEM) for microstructural features of their surfaces, as described elsewhere [42,50].

### 2.8. Transcription Profiling

The cultures of each strain were initiated by spreading 100 μL aliquots of a 10^7^ conidia/mL suspension on cellophane-overlaid SDAY plates or shaking 50 mL aliquots of a 10^6^ conidia/mL suspension in TPB and incubated for 7 days under the optimal regime or 5 days at 25 °C. Total RNAs were extracted from the 3-, 5-, and 7-day-old SDAY cultures or 3-, 4-, and 5-day-old TPB cultures with the RNAiso Plus Kit (TaKaRa, Dalian, China), and reversely transcribed into cDNAs with the PrimeScript^RT^ reagent kit (TaKaRa), respectively. Transcripts of *csn5*, 57 E1, E2, E3, and USP (*uspA*-like) genes; 3 activator genes (*brlA*, *abaA*, and *wetA*) of the central developmental pathway (CDP) plus downstream *vosA*; 2 key hydrophobin genes (*hyd1* and *hyd2*); and 10 heat responsive genes were quantified from the cDNA samples derived from three independent cultures of each strain via qPCR with paired primers (Appendix A) under the action of SYBR Premix Ex Taq (TaKaRa). The fungal β-actin gene was used as a reference. A threshold cycle (2^−∆∆CT^) method was used to compute relative transcript levels of each gene in the *csn5* mutants with respect to the WT standard. One-fold transcript change was used for the significance of each gene downregulated or upregulated in the absence of *csn5*.

## 3. Results

### 3.1. Sequence Comparison of Fungal Csn5 Orthologues

Csn5 orthologues found in the NCBI databases of entomopathogenic and nonentomopathogenic fungi with the query of *A. nidulans* CsnE (Q5BBF1) were clustered into distinct clades consistent with host lineages in phylogeny and shared homology in structure (Appendix A). Like those orthologous to the query, *B. bassiana* Csn5 (EJP66879, 338 amino acids (aa)) fell into the Mov34/MPN/PAD-1 family, sharing the conserved JAMM motif involved in zinc ion coordination and the active site for isopeptidase activity, and featured N-terminal JAB_MPN and C-terminal CSN5_C domains. An NLS motif appeared at the N- or C-termini of most Csn5 orthologues with predicted scores of 2.0–2.5 (*Aspergillus*, *Alternaria*, *Trichoderma*, and *Ustilago*) to 7.0 (*Metarhizium*). The NLS scores were intermediate (3.1–3.5) for Csn5 orthologues in Cordyceptitaceae but unpredictable for those in *N.crassa*, *Botrytis cinerea**,* and *Pyricularia oryzae*. Therefore, variable NLS scores implied that the roles of Csn5 in nuclear events might vary with fungal lineages.

Aside from Csn5, *B. bassiana* had six other putative CSN subunits orthologous to those studied in *A. nidulans* [30,31,36] including Csn1–4, Csn6, and Csn7 (Appendix A).

### 3.2. Subcellular Localisation of Csn5 and Its Essentiality for Ubiquitination

A nuclear localisation of Csn5, implicated by its NLS that was predicted at a score of 3.5, was clarified using the LSCM images of the green fluorescence-tagged Csn5 fusion protein expressed in the WT strain. The fusion protein accumulated more heavily in the nuclei than in the cytoplasm of hyphal cells stained with the nucleus-specific dye DAPI, shown in red, which overlapped well with the expressed green fluorescence in the nuclei (Figure 1A). The N/C-GFI ratios measured from the stained hyphae reached 2.34 on average (Figure 1B). More accumulation of Csn5 in the nucleus confirmed its involvement in nuclear events such as transcriptional mediation [16].

Due to the role of CSN in ubiquitination/deubiquitination [13] and the confirmed interaction of CsnE/Csn5 with the deubiquitinase UspA in *A. nidulans* [37], the role of Csn5 in ubiquitination was analysed using the hyphal protein extracts of each strain. Despite similar β-actin blots, western blots probed with the rabbit monoclonal anti-ubiquitin antibody showed a 68% increase in free ubiquitin accumulation in the extracts of ∆*csn5* compared to the control (complementation and WT) strains (Appendix A). Next, transcript levels of 57 putative UPS genes (Appendix A) were assessed in the cDNA samples derived from the 3-day-old SDAY cultures. Among 3 E1 and 14 E2 genes, the expression levels of 1 E1 (*ubi3*) and 8 E2 genes were reduced by 50–92% in ∆*csn5* relative to WT, accompanied by significant upregulation of *ubi4* (E1 gene) and 3 other E2 genes (Appendix A). Among 23 E3 genes, however, only a few were moderately dysregulated in ∆*csn5* (Appendix A). Among 17 USP genes analysed, interestingly, 2 (*usp1* and *usp5*) were abolished at the transcriptional level, but none were upregulated in ∆*csn5* (Appendix A). These data demonstrated an active role of Csn5 in mediating expression levels of some UPS genes required for a balance of ubiquitination/deubiquitination.

### 3.3. Indispensability of Csn5 for Fungal Insect Pathogenicity

In the standardised bioassays, the control strains killed all *G. mellonella* larvae within 10 days after NCI or within 6 days after CBI (Figure 2A). In contrast, the ∆*csn5* mutant killed very few of the tested larvae via NCI and showed slower kill action via CBI, resulting in the inability to estimate LT_50_ via NCI and an LT_50_ prolonged by 17% via CBI in comparison to the WT estimate of 3.4 days (Figure 2B). These data indicated the indispensability of Csn5 for the fungal NCI and pathogenicity and its involvement in certain cellular events during haemocoel colonisation post-infection.

For insight into the nearly abolished pathogenicity, the conidial adherence crucial for NCI initiation was assessed on locust hind wings. As a result, conidial adherence to the wing cuticle was reduced by 67% in ∆*csn5* relative to WT (Figure 2C,D). Next, the CBI-killed larvae were incubated at 25 °C to determine whether intrahaemocoel hyphae penetrated the insect cuticle for outgrowth, which would also be indicative of hyphal invasion into the insect body for successful NCI. Compared to heavy outgrowths of the control strains covering all cadavers 4 days post-death, the outgrowth of ∆*csn5* was rare on the cadaver surfaces (Figure 2E), implicating an impaired capability for its penetration through the cuticle. This implication was verified by the reduced activities of the ECEs and Pr1 proteases required for cuticle degradation during NCI [43,45]. After the 3-day shaking incubation of a 10^6^ conidial/mL suspension in CDB-BSA, total ECEs and Pr1 activities were reduced by 98.4% and 99.7% in the supernatants of the ∆*csn5* cultures in comparison to the WT counterparts (Figure 2F). The reductions indicated the abolished secretion of various cuticle-degrading enzymes required for the NCI of ∆*csn5*.

Further, haemolymph samples taken from surviving larvae post-NCI or post-CBI were examined under a microscope to reveal whether the ∆*csn5* mutant was successful in haemocoel colonisation post-NCI or post-BCI. At 84 h post-CBI or 100 h post-NCI, control strains formed abundant hyphal bodies (i.e., blastospores), enabling proliferation by yeast-like budding for the acceleration of mycosis development and host death (Figure 2G). For ∆*csn5*, however, hyphal bodies were absent even at 144 h post-NCI and did not appear sporadically in the samples until 112 h post-CBI, implying a blockage of the dimorphic (hypha–blastospore and vice versa) transition required for haemocoel colonisation post-NCI and hyphal outgrowth post-death. For further insight into delayed dimorphic transition via CBI, hydrocarbon epitopes on the surfaces of conidia, which comprise pathogen-associated molecule patterns (PAMPs) perceived by host PAMP recognition receptors and associated with fungal response to host immune defence [51], were examined in fluorescent lectin assays. As a result, hydrocarbon epitope patterns on the surfaces of the ∆*csn5* conidia were altered, with 47% more α-glucose and α-N-ace- tylglucosamine (GlcNAc) residues labelled by ConA and 58% more β-galactose residues labelled by PNA than the WT counterparts (Figure 2H). The 3-day shaking incubation of a 10^6^ conidia/mL suspension in TPB, mimicking insect haemolymph, resulted in a dimorphic transition rate decreased by 56% in ∆*csn5* despite little biomass difference between the mutant and control strains (Figure 2I). The mutant blastospores were also compromised in the cell cycle, with shortened G1 and prolonged S and G2 phases (Figure 2J), and in the hydrocarbon epitope pattern, with 30% more β-galactose residues labelled by PNA, 21% more mannose residues labelled by *Galanthus nivalis* lectin (GNL), and 30% reduced β-GlcNAc and sialic acid residues labelled by wheat germ agglutinin (WGA) (Figure 2K).

Altogether, Csn5 was indispensable for fungal pathogenicity via NCI. The indispensability was due to its essentiality in both conidial hydrophobicity and adherence to insect cuticle and the secretion of cuticle-degrading enzymes required for NCI, aside from its role in the post-infection cellular events associated with haemocoel colonisation and lethal action.

### 3.4. Roles of Csn5 in Radial Growth and Stress Response

Hyphal growth and response to stress cues generated from host immune defence are crucial for hyphal invasion into the host body via NCI. Compared to the control strains, the ∆*csn5* mutant showed moderate growth defects on rich SDAY and minimal CDA based on the morphology and sizes of the colonies grown for 8 days at optimal 25 °C after initiation with 10^3^ conidia (Figure 3A). Interestingly, the mutant was hypersensitive to a 3-h heat shock at 42 °C during an 8-day incubation on SDAY at 25 °C as well as sensitive to two oxidants (H_2_O_2_ and menadione) in CDA despite null responses to osmotic (NaCl and sorbitol) and cell wall-perturbing agents (Congo red and calcofluor white). Based on percent changes of relative growth inhibition, the ∆*csn5* mutant became 26% more sensitive to H_2_O_2_ and 33% more sensitive to the heat shock (Figure 3B) to which 2-day-old SDAY colonies were exposed for subsequent 6-day growth recovery at 25 °C. The mutant’s increased sensitivity to menadione was significant but moderate. The increased sensitivities of the mutant to the two oxidants correlated with 29% and 28% reductions in the total activities of the SODs and catalases (Figure 3C) required for the decomposition of superoxide anions and H_2_O_2_ in *B. bassiana* [52,53,54], respectively. Moreover, its hypersensitivity to the heat shock correlated with 5 of 10 heat-responsive genes [46,55,56] markedly downregulated by 50–79% and 2 others less downregulated (Figure 3D).

All these changes were well restored in the complementation strain, and hence, indicated the active role of Csn5 in heat shock and the antioxidant responses of *B. bassiana*, despite its dispensability for radial growth under normal conditions and osmotic and cell wall-perturbing stresses.

### 3.5. Essential Roles of Csn5 in Asexual Development and Cell Hydrophobiciy

Aerial conidiation and conidial maturation are asexual developmental processes that are crucial for fungal survival/dispersal in host habitats and are genetically governed by three CDP activator genes and downstream *vosA* in *B. bassiana* [57,58]. The WT strain usually starts conidiation as early as day 3 after initiation of SDAY cultures through spreading 100 μL aliquots of a 10^7^ conidia/mL suspension and achieves maximal conidial yield on day 7 or 8 [48,49]. In this study, zigzag rachises (conidiophores) and conidia were observed in the 3-day-old SDAY cultures of control strains under the optimal regime, contrasting with differentiated ∆*csn5* hyphae, hardly observed until day 6 (Figure 4A). In ∆*csn5*, the key CDP genes *brlA* and *abaA* were downregulated, respectively, by 75% and 62% on day 3 and 97% and 73% on day 5, at the time of which the expression of the *wetA* or *vosA* required for conidial maturation [57] was also suppressed or nearly abolished (Figure 4B). As a consequence, conidial yields decreased sharply by 85%, 76%, and 70% in the 5-, 7-, and 9-day-old cultures of ∆*csn5* relative to WT, respectively, although biomass accumulation levels showed little variability (*p* > 0.05) among the cultures of all tested strains (Figure 4C). In addition, the quality of the ∆*csn5* conidia was compromised by a 43% decrease in hydrophobicity, a 29% increase in GT_50_ at 25 °C, and a 16% reduction in LD_50_, indicative of UVB resistance (Figure 4D).

Previously, class I and II hydrophobin genes (*hyd1* and *hyd2*) in *B. bassiana* proved essential for hydrophobin biosynthesis and assembly into rodlet bundles of the conidial coat determinant to hydrophobicity and adherence to the insect cuticle [59]. The decrease of conidial hydrophobicity in ∆*csn5* was similar to a decrease caused previously by double deletion of *hyd1* and *hyd2*, suggesting the involvement of *csn5* in the transcriptional mediation of *hyd1* and *hyd2*. This speculation was verified bytheconcurrence of abolished *hyd2* expression with 43% and 68% downregulation of *hyd1* in the 3- and 5-day-old SDAY cultures of ∆*csn5* relative to WT (Figure 4E). During the 72-h monitoring, moreover, the dew-like water droplets showed little morphological change on the cultures of either control strain but gradually sank into and eventually disappeared on the surface of the ∆*csn5* culture (Figure 4F), a phenomenon indicative of more hydrophilic traits for the mutant culture. SEM images of conidial surfaces displayed an outermost layer of well-defined rodlet bundles for the control strains (Figure 4G). In contrast, such rodlet bundles were severely impaired, becoming barely distinguishable residues on the surfaces of the ∆*csn5* conidia, which resembled a ‘bald’ phenotype caused by the deletion of *hyd1* or the double deletion of *hyd1* and *hyd2* in the previous study [59].

Taken together with all phenotypic and transcriptional changes observed in ∆*csn5* and restored by targeted gene complementation, Csn5 was essential for asexual development and cellular hydrophobicity due to its role in regulating the key CDP and hydrophobin genes.

## 4. Discussion

CsnE/Csn5 can interact with UspA in *A. nidulans* [37]. Due to its role in transcriptional mediation [16] and the large number of Csn5-regulated genes identified from the transcriptomes of *A. alternata* and *P. fici* [39,40], our study focused on the possible role of Csn5 in the expressions of UPS genes as well as those required for asexual development; hydrophobin biosynthesis and stress response; and the quality/function control of those enzymes crucial for conidial adherence, cuticular penetration, and antioxidant activity. Our data suggest the essentiality of Csn5 for the ubiquitination/deubiquitination balance of *B. bassiana*. The suggested essentiality was uncovered by both increased ubiquitin accumulation and 18 dysregulated E1, E2, E3 and USP genes in ∆*csn5*. Thirteen of these genes were sharply repressed or abolished at the transcriptional level, although almost all of them remained functionally unknown. In ∆*csn5*, none of the 17 analysed USP genes were upregulated aside from the abolished expression of *usp1* and *usp5*. This can be distinguished from all analysed USP genes (*uspA*–*G*) upregulated in *A. nidulans* ∆*csnE* [37]. Previously, free ubiquitin accumulation was abolished through the deletion of *ubi4*, an E1 gene unlike *ubi1/2* or *ubi3* required for cell viability [7], and increased through the deletion of E3 *ubr1* (i.e., *ubl7* in Appendix A) in *B. bassiana* [8], as seen in *A. nidulans* ∆*uspA* [40]. Repressed *ubi3* and upregulated*ubi4* imply that their effects on ubiquitination could be counteracted as a result of *csn5* disruption. The role of deleted *uspA* in the *A. nidulans* deubiquitination suggests similar roles for the two abolished USP genes in our ∆*csn5* mutant. We infer that the increased ubiquitin accumulation in ∆*csn5* could have resulted from 11 dysregulated E2 or 2 abolished USP genes or, less likely, from 2 less downregulated E3 genes (*upl1* and *upl4*), the effects of which could be counteracted by other E3 genes differentially upregulated. Therefore, Csn5 is involved in the normal expression of some UPS genes and the balanced ubiquitination/deubiquitination that may mediate the expression of crucial phenotype-related genes as well as the quality control of those enzymes essential for or influential on the fungal lifecycle, as discussed below.

The indispensability of Csn5 for insect pathogenicity relies upon its regulatory role in both the expression of two *hyd* genes determinant of conidial hydrophobicity and adherence to the insect cuticle as well as the secretion of extracellular enzymes required for NCI. In ∆*csn5*, the repressed/abolished expressions of *hyd1* and *hyd2* required for hydrophobin biosynthesis and assembly into rodlet bundles of conidial surfaces [59] apparently led to severely impaired conidial coats and marked reductions in conidial hydrophobicity and adherence to locust wing cuticles. This is also supported by conidial hydrophobicity and cuticle adherence tied to adhesin 2 (Adh2), which functions uniquely in host infection among the three adhesins characterised in *B. bassiana* [43]. The reduced adherence implicated that two-thirds of topically applied conidia failed to initiate NCI through adhesion to the insect cuticle. The hyphae formed after germination of the remaining conidia attached to the insect cuticle lost the capability for secreting the cuticle degrading ECEs and Pr1 proteases required for successful NCI [44,45]. For ∆*csn5*, the lost capability was also implicated by the blocked outgrowth of its intrahaemocoel hyphae on the cadaver surfaces of those larvae dead post-CBI. Apparently, Csn5 played an essential role in the secretion of various cuticle-degrading enzymes, which could have been out of control in terms of quality and activity when *csn5* lost function. Thus, both the reduced hydrophobicity/adherence of conidia and the abolished secretion of those NCI-aiding enzymes were responsible for the abolished pathogenicity of ∆*csn5* via NCI. The profound effect of *csn5* on fungal insect pathogenicity was similar to that observed in *A. alternata*, which lost plant pathogenicity when *csn5* was deleted [40]. This reinforces the essentiality of Csn5 for fungal pathogenicity in different hosts.

Moreover, Csn5 participates in transcriptional mediation of *brlA* and *abaA*, two key CDP genes that regulate the aerial conidiation and submerged blastospore production indicative of dimorphic transition in vivo in *B. bassiana* [58]. Despite moderate defects in radial growth and unaffected biomass accumulation in the cultures, our ∆*csn5* mutant showed at least 70% reduction in its aerial conidiation level during normal incubation. This severe conidiation defect can be distinguished from abolished conidiation in *P. fici* ∆*csnE* [39] or *A. alternata* ∆*csn5* [40]. In this study, the blocked ∆*csn5* proliferation in vivo was associated with attenuated virulence via CBI and evidenced by a dimorphic transition rate lowered by 56% in the TPB cultures mimicking insect haemolymph. The compromised conidiation and blastospore production coincided well with sharply reduced expression levels of both *brlA* and *abaA* in the ∆*csn5* cultures, indicating a critical role for Csn5 in activating either key CDP gene required for aerial conidiation and submerged blastospore production [58]. Aside from the influences of repressed *brlA* and *abaA* on asexual development, hyphal differentiation and development could be affected by the decreased hyphal hydrophobicity containing more water in the ∆*csn5* cultures, in which dysfunctional *hyd1* and *hyd2* could impede the biosynthesis and assembly of the hydrophobins required for aerial hyphal growth and differentiation [60]. The blocked proliferation in vivo of ∆*csn5* post-CBI also could be partially due to its being less capable of collapsing the host immune defence, which might generate reactive oxygen species for scavenging by SODs and catalases [3] and the host fever [61] required for fungal adaptation through induced expression of heat-responsive genes [46,55,56]. Both the reduced activities of antioxidant enzymes and the repressed expression of several heat-responsive genes in the ∆*csn5* cultures implicate an active role of Csn5 in sustaining the functions of antioxidant enzymes and heat-responsive genes in *B. bassiana*.

## 5. Conclusions

Csn5 is required for insect pathogenicity, cellular hydrophobicity, and asexual development and is also functional in the cellular response to oxidative stress and heat shock in *B. bassiana*. Our findings unravel that the indispensability of Csn5 for fungal insect pathogenicity via NCI relies upon its prominent role in regulating the expression of the *hyd1* and *hyd2* essential for the conidial hydrophobicity and cuticle adherence that initiate NCI as well as the secretion of cuticle-degrading enzymes required for successful NCI. The essentiality of Csn5 for aerial conidiation and submerged blastospore production associated with haemocoel colonisation post-infection depends on its role in the transcriptional coordination of *brlA* and *abaA* serving as key CDP activators. These findings offer novel insight into the significance of Csn5 for the fungal lifecycle in vivo and in vitro and implicate that the Csn5-dependent ubiquitination/deubiquitination balance regulates phenotypes essential for the biological control potential of *B. bassiana* against arthropod pests.

## Figures and Tables

**Figure 1 jof-07-00642-f001:**
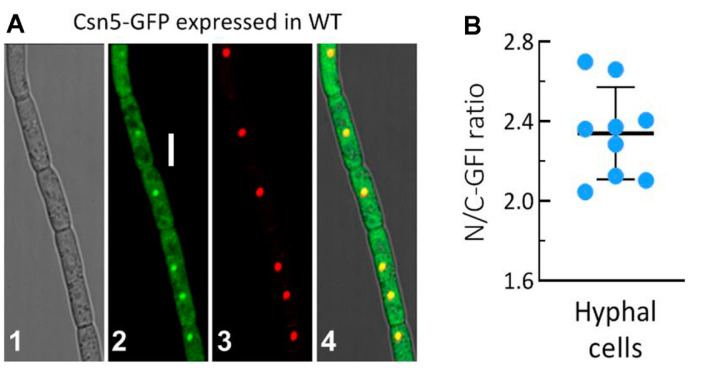
Subcellular localisation of Csn5 in *B. bassiana*. (**A**) LSCM images (scale bar: 5 μm) for subcellular localisation of the Csn5–GFP fusion protein in the hyphal cells stained with DAPI nuclear dye (shown in red) after collection from 2-day-old SDBY culture. Panels 1, 2, 3, and 4 are bright, expressed, stained, and merged views of the same field. (**B**) Nuclear versus cytoplasmic green fluorescence intensity (N/C-GFI) ratios of the fusion protein measured from in the cells of nine hyphae.

**Figure 2 jof-07-00642-f002:**
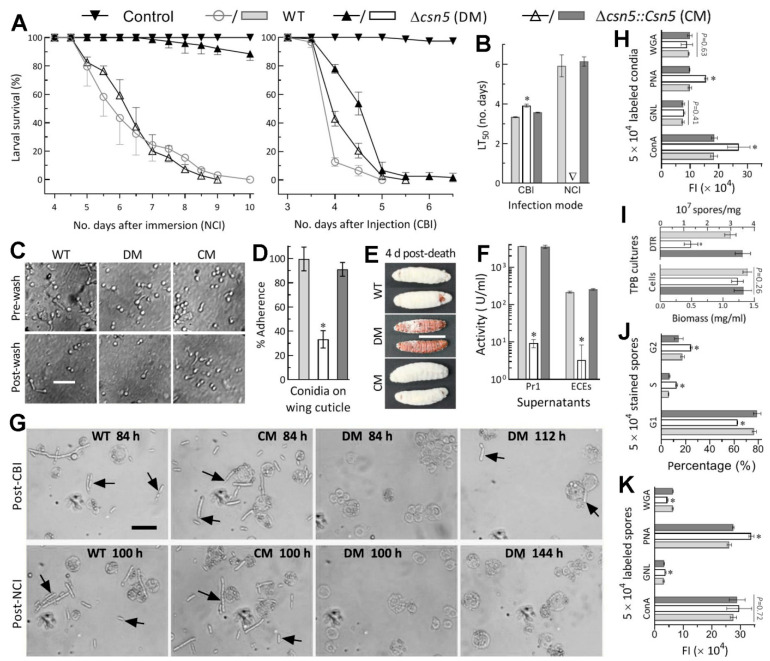
Essential role of Csn5 in the infection cycle and related cellular events of *B. bassiana*. (**A**,**B**) Time–survival trends of *G. mellonella* larvae after topical application (immersion) of a 10^7^ conidia/mL suspension for normal cuticle infection (NCI) and intrahaemocoel injection of ~500 conidia per larva for cuticle-bypassing infection (CBI) and LT_50_s (number of days) estimated from the trends. (**C**,**D**) Microscopic images (scale bar: 20 μm) for conidia attached to locust hind wings pre-wash and post-wash and conidial adherence assessed as percent ratios of pre-wash counts over post-wash counts with respect to the WT standard, respectively. (**E**) Images (scale bar: 10mm) of hyphal outgrowths on the surfaces of insect cadavers 4 days post-death via CBI. (**F**) Total activities of cuticle-degrading ECEs and Pr1 proteases quantified from the supernatants of 3-day-old CDB-BSA cultures, which were initiated by shaking 10^6^ conidia/mL suspensions at 25 °C. (**G**) Microscopic images (scale bar: 20 μm) for the presence and abundance of hyphal bodies (arrowed) and host haemocytes (spherical or subspherical cells) in haemolymph samples taken from surviving larvae post-NCI or post-CBI. Note that the ∆*csn5* mutant failed to produce hyphal bodies 144 h post-NCI and produced very few 110 h post-CBI. (**H**) Carbohydrate epitope patterns of conidia labelled with four fluorescence lectins. (**I**) Biomass levels and dimorphic transition rates measured from 3-day-old TBP cultures mimicking insect haemolymphs. (**J**,**K**) Distribution of DNA profiles for the cell cycle phases of DNA-stained blastospores in the TPB cultures and carbohydrate epitope patterns of those blastospores labelled with four fluorescence lectins, respectively. * *p* < 0.05 (**B**), * *p* < 0.01 (**H**,**J**,**K**), or * *p* < 0.001 (**D**,**F**,**I**). Error bars: SDs of the means from three independent replicates.

**Figure 3 jof-07-00642-f003:**
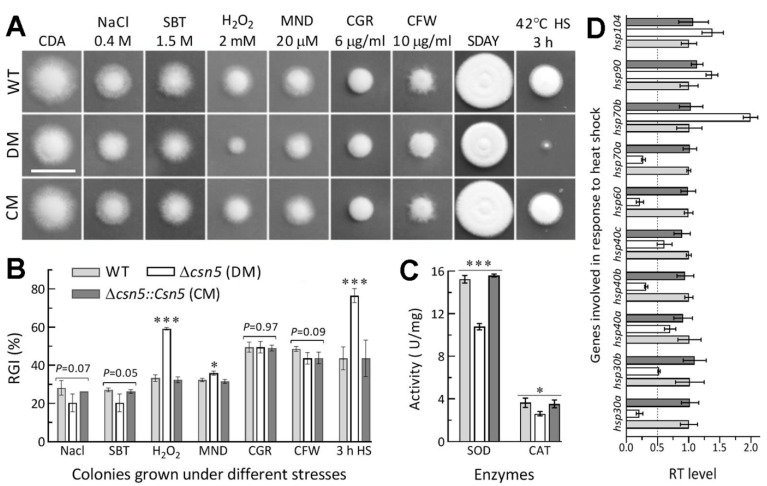
Impacts of *csn5* disruption on radial growth under normal and stressful conditions. (**A**) Images (scale bar: 2 cm) for 8-day-old colonies grown at 25 °C on rich SDAY alone (control) or exposed to a 3-h heat shock at 42 °C after 2-day growth as well as minimal CDA alone (control) or supplemented with the indicated concentrations of chemical stressors (SBT, sorbitol; MND, menadione; CGR, Congo red; CFW, calcofluor white). (**B**) Relative growth inhibition (RGI) percentages of fungal colonies under the stressors. All colonies were initiated by spotting 1 μL aliquots of a 10^6^ conidia/mL suspension. (**C**) Total activities of superoxide dismutases (SOD) and catalases (CAT) quantified from the protein extracts of 3-day-old SDAY cultures. (**D**) Relative transcript (RT) levels of heat-responsive genes in the 3-day-old SDAY cultures of *csn5* mutants with respect to the WT standard. The dashed line indicates a significant level of one-fold (50%) downregulation. * *p* < 0.05 or *** *p* < 0.001 in Tukey’s HSD tests. Error bars: SDs of the means from three independent replicates.

**Figure 4 jof-07-00642-f004:**
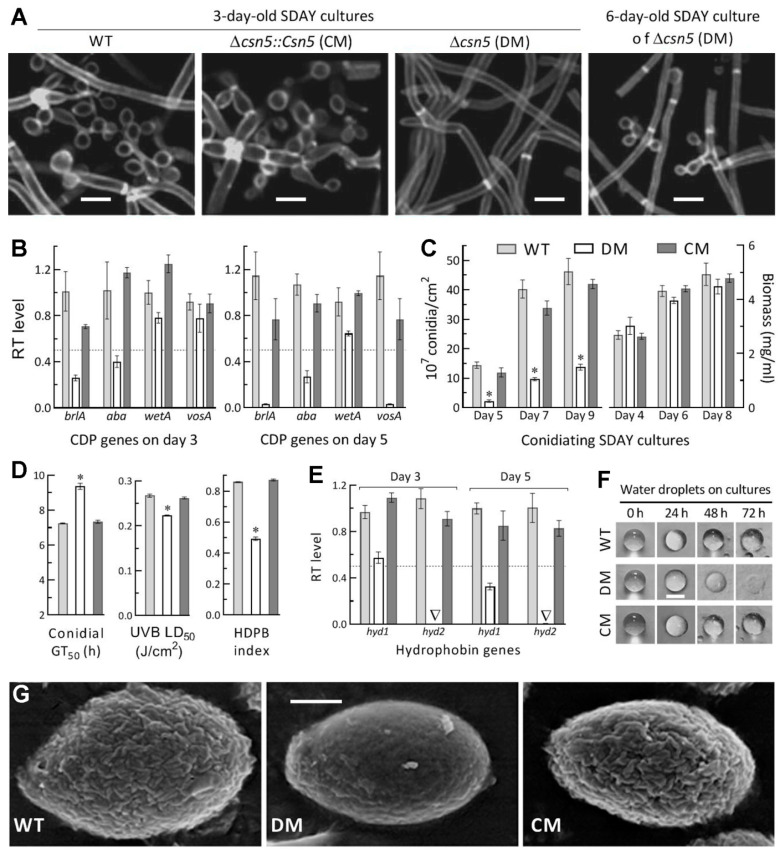
Essential role of Csn5 in the mediation of aerial conidiation and conidial quality. (**A**) Microscopic images of the conidiation status of samples taken from 3- and 6-day-old SDAY cultures and stained with the cell wall-specific dye, calcofluor white. (**B**) Relative transcript (RT) levels of three CDP genes and downstream *vosA* in the 3- and 5-day-old SDAY cultures of *csn5* mutants with respect to the WT standard. (**C**) Conidial yields and biomass levels quantified from the SDAY cultures during a 9-day incubation under the optimal regime of 25 °C and L:D 12:12. (**D**) The indices of conidial quality presented by GT_50_ for 50% of conidial germination at 25 °C, the hydrophobicity (HDPB) index assessed in an aqueous–organic system, and LD_50_ for resistance to UVB irradiation. (**E**) RT levels of *hyd1* and *hyd2* in the 3- and 5-day-old SDAY cultures of *csn5* mutants with respect to the WT standard. (**F**) Dispersal of dew-like water droplets on the surfaces of 4-day-old SDAY cultures. (**G**) SEM images (scale bar: 0.5 μm) for microstructures of the outermost hydrophobin rodlet bundles on conidial surfaces. All SDAY cultures were initiated by spreading 100 μL of a 10^7^ conidia/mL suspension per plate. The dashed line (**B**,**E**) indicates a significant level of one-fold (50%) downregulation. * *p* < 0.001 in Tukey’s HSD tests. Error bars: SDs of the means from three independent replicates.

## Data Availability

All data presented in this study are included in the paper and associated supplementary materials.

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
