# Peer review of "Essential Role of COP9 Signalosome Subunit 5 (Csn5) in Insect Pathogenicity and Asexual Development of Beauveria bassiana"

_jof, 2021, doi:10.3390/jof7080642_

Round 1

Reviewer 1 Report

Review for Mou et al.

The submitted manuscript by Mou et al. presents a phenotype for the Ko of CSN5 in the insect pathogenic typical filamentous fungal species, Beauveria bassiana, of the hypocrealean genera, which has a high potential for insect control. The authors show a nice link between a csn5 deletion mutant strain in the fungi, and a decrease in fungal pathogeny. The manuscript includes 4 figures: (i) cellular localization of CSN5 and the possible link with ubiquitination; (ii) morphological phenotypes of the mutants upon infection;(iii) evaluating the effect of cellular stresses on CSN mutants; (iv) a try to bond between the loss of Csn5 and the expression of certain genes involved in conidiation.

The novelty of the manuscript is with the morphological phenotypes, especially, the SEM ultrastructural image of the conidia and the related hydrophobicity phenotypes. However, this phenotype is literally not explained. Also, the authors made various observations without linking them with each other, or deeply researching the meaning of them and how they may link with each other. This reviewer suggests publishing the manuscript only after a major revision, which will include (1) changes in the text, (2) adding more references, (3) adding a descriptive figure as figure 1 (explained below), (4) changing the title, (5) several experiments in order to strength the data. Please find below the description of major and minor comments.

Title

The current title does not describe the novelty of the paper. The manuscript does not show a clear link between the ubiquitin-proteasome system (UPS) and the pathogenicity. Figure 1 shows a (known and deeply studied) link between CSN5 and the UPS. Thus, the title could be – “The loss of CSN5 in the filamentous fungi Beauveria bassiana abolished insect pathogenicity”. The title cannot claim for something that is not shown in the manuscript.

Abstract

  • Please largely underestimate the UPS in the abstract. The manuscript is not on the UPS, and as written above, the authors do not show a direct link between their data and the UPS.
  • Line 13 – “COP9 signalosome (CSN) subunit 5 (Csn5) can activate CSN deNEDDylase” CSN5 is not an “activator” of the CSN but the deneddylase subunit of the CSN complex through a metalloprotease motif.

Introduction

  • Line 40 - “ubiquitin-proteasome (UbP) pathway” – please use UPS, which is the common terminology instead.
  • Line 45 – “COP9” is initial for “constitutive morphogenesis number 9”, indeed from plants. “CSN” is initial for “COP9 signalosome”, which links together the plant complex (COP9) and the mammalian complex “signalosome”. Please cite properly also Seeger M et al. 1998.
  • Line 47 – for deubiquitination, please add a reference by Groisman et al. 2003.
  • The authors miss a very important issue. In all organisms (conserved from human and plants to yeast), CSN5 is found in a complex context, but also in a free form. Which one of the forms is important in the case of the submitted data? It is not shown, however since the manuscript is on CSN5, the authors should describe the free form as well (DOI: 1074/jbc.M113.468959 ).
  • Line 49-50 – please rephrase.  
  • Line 52 – “ubiquitin-like neddylase” NEDD8 is a neddylase????? .. please correct.
  • Line 55 – As you describe this organism for the first time in the csn5 context, please add references for more fungal species CSN and describe their phenotypes (S. pombe and cell cycle (DOI: 1091/mbc.01-10-0521 ), Candida albicans ( DOI: 10.3389/fmicb.2016.00401 and DOI: 10.1128/EC.05250-11) , S. cerevisiae and ergosterol DOI: 10.1096/fj.201902487R ) or mating (DOI: 10.1093/embo-reports/kvf235 ), Neurospora crassa and the circadian clock  (DOI: 10.1101/gad.1322205) etc.
  • Line 60 – “role of CSN in mediating cullin neddylation”…. Not neddylation but vice versa!
  • Line 61 – “mutations of JAMM (JAB1/MPN/Mov34 61 metalloenzyme)”. Don’t put the cart before the horse… Please explain somewhere above CSN5 composition, activity, domains, and only then explain about the mutations.
  • Line 63 - “without interference with 62 CSN assembly or CSN-cullin interactions” – please rephrase.
  • Line 70 – should come earlier.
  • Line 86 – please cite also (DOI: 3390/biom10071082 )

Materials and methods

  • For 2.1 I suggest to add a figure and replace it (partially with figure 1). The figure will contain Fig 1a, a PDB model of B. bassiana on the human CSN5, a phylogenomic tree, alignment with the closest homologues compared to Arabidopsis or human (neurospora, Aspargillus etc..). Similarly, alignment for the NLS.
  • Line 117 - 116 (http://blast.ncbi.nlm.nih.gov/blast.cgi/). The link leads to an error. Please correct.
  • Line 123 – the signature of “::” usually means a replacement and you actually used it for a chimera. Please correct all over the manuscript: Csn5::GFP = replacement; Csn5-GFP = chimera.
  • Line 155 – make sure the way you write gene, protein and mutation is correct all over the manuscript (delta csn5::csn5 should be delta csn5::CSN5)
  • Line 157 – font
  • Line 178 – (0.02% Tween) do you mean that this is the concentration of tween 80 in the other samples, thus you have used it as a control?
  • Line 179 – language - at optimal TEMPERATURE OF 25C
  • Line 182 – LT50 – please explain
  • Line 192 – please phrase
  • Line 201 – which microscope you have used? Brand? Magnification?
  • Line 215-216 – why night/day? What is the phylogenetic of this fungi and what is the benefit?

Results:

Major comments – The authors conclusions do not suit with the experiments they were chosen. For example: they have used RT PCR, why not checking whether the expression of CSN5 changes the fate of other CSN subunits? What about some well-known substrates of CSN in filamentous fungal species (which could be found in manuscripts by the He or Braus labs). The whole concept of figure 1 is wrong. The authors needed first to figure out if CSN5 is required in a complex context for example by repeating the experiments in another CSN mutant (such as CSN6) and evaluate if getting similar results; as well as by treating WT with the inhibitor csn5i-3 instead of the mutant, and assessing if receiving the same phenotype. This is important since the metalloprotease is active only in a complex context. Without effect of the inhibitor, they might deal with the free form of the CSN. They could also repeat experiments in WT pretreated by MLN4924, to see whether they get a similar phenotype as the mutant, this would suggest that they deal with CSN5 regulation of cullins. Both would suggest the involvement of the ubiquitin pathway. Other inhibitor to be used is the proteasome inhibitor MG132… At the moment they cannot claim for regulation of the phenotype by the UPS. Another required experiment is to provide more data on the genetic interaction of the hyd1/2 genes and CSN5 as written below.  

I suggest one out of 2 –

  • The change the manuscript to a more morphological one and take of the ubiquitin part in fig 1 (or remove it to the supplementary materials)
  • Or – repeat several of the experiments with the abovementioned inhibitors to assess if the story is linked with the UPS.

The figures:

Fig 1 –

Overall the figure does not confirm a link with the UPS, but that CSN5 regulates the UPS. It could be interesting to learn why, but the information given in Figure 1 is not explained, not even the names of the enzymes are written in some of the illustrations and they appear as numbers. We do not even get information on which of the enzymes expression had altered and which did not (but only a number). And in any case, the results of the illustration do not add information or a link between the pathogenicity and the UPS. Figure 1C is not publishable and at any case it is nonsense – is this mono-ubiquitin? What about ubiquitin chains? I suggest to leave only A-B in this figure and to transfer the rest to the “supplementary materials.

Line 269 – terminology of CSN5 domains is wrong. Please rephrase.

Line 273 – “zeros” – please phrase.

Line 277 – In most organisms (very few examples - yeast - DOI: 10.1093/embo-reports/kvf235  ; planta DOI: 10.1105/tpc.8.11.2047 ; S. pombe doi: 10.1016/s0960-9822(00)80091-3). Moreover, the role of the CSN in transcription was earlier summarized -  ·  DOI: 10.1038/embor.2009.33 .

Line 280 – “nucleus-specific dye” please add details.

Lines 296-297 – neither the figure or the description claims to specific enzymes’ names.

Line 284 – please rephrase “More accumulation of Csn5 in nucleus suggests its possible role in nuclear events, such as transcriptional mediation”. instead of “suggests” please write “confirm” as was suggested before and summarized at Chamovitz 2009 DOI: 10.1038/embor.2009.33 .

Figure 2

For Figures 2C, E, G please indicate (either on the figure or in figure legends), which of the picture describes delta csn5 (not enough to write DM/CM). Similarly, I figure 3A.

Figure 4

The data in figure 4 is the most important, both the reduced expression of hyd1 and hyd2 and the phenotype, however more information is required. At least show the phenotype of the double mutant (hyd1-2 and csn5) and the phenotype of single mutants as for control to confirm the genetic interaction and complete the missing link.  

Author Response

.

Reviewer 2 Report

Authors have carried out a fine piece of research in which they supply data to prove, very elegantly, the role of ubiquitin cycle in the pathogenicity and asexual reproduction of the entomopathogenic fungus B. bassiana (Bb). The manuscript is very well written and the figures clearly show the points authors want to make. My sincere congratulations. I only have a few points I want to raise, which they may consider for completing their explanation to the facts the csn5 mutant has raised.

-The first is the name of the mutant, csn has been recently used for describing chitosanase genes in fungi (See papers by Aranda-Martinez et al and by Suarez-Fernandez et al). Can they rename their genes?

-I miss in the paper some implications of the present work for the practical usage of the fungus

-p.4l.192.  "larvae died post-CBI" should read"larvae dead post-CBI"

-p.5l.198-199. Why authors did not use insect cuticle as enzyme inducer for Bb?

-p.6l.16 I have always used disregulation but it seems that dysregulation is also correct

-Fig.1E. Do authors have any explanation why ubi4 ubic3 ubc5 ubc 12 are up-regulated in deltacsn5?. Same for Fig1F in E#Ub-ligase genes6 and 15?

-Authors do not indicate (or I have not found it) specificity for PNA lectin

-Fig. 2A why has the complemented mutant impaired pathogenicity?. The ECE abbreviation is not explained fully

-Fig3G I would not use ultrastucture for SEM data (I would for TEM). Use SEM surface microstructure of conidia instead

Author Response

Comments and Suggestions for Authors

Authors have carried out a fine piece of research in which they supply data to prove, very elegantly, the role of ubiquitin cycle in the pathogenicity and asexual reproduction of the entomopathogenic fungus B. bassiana (Bb). The manuscript is very well written and the figures clearly show the points authors want to make. My sincere congratulations. I only have a few points I want to raise, which they may consider for completing their explanation to the facts the csn5 mutant has raised.

Author response: Thanks a lot for understanding and encouragement.

-The first is the name of the mutant, csn has been recently used for describing chitosanase genes in fungi (See papers by Aranda-Martinez et al and by Suarez-Fernandez et al). Can they rename their genes?

Author response: Thanks. I have found the paper published by Aranda-Martinez et al. (Environ Microbiol 18: 4200-4215, 2016) but not that by Suarez-Fernandez et al. Aranda-Martinez et al. (2016) identified a long list of carbohydrate-active enzymes in the genome of the nematophagous fungus Pochonia chlamydosporia, including chitosanases mentioned as CSN subunits (Csn1-11). This is conflicting with no more than eight CSN subunits (Csn1-8/CsnA-H) existing in model fungi. I tried to compare the sequence of P. chlamydosporia Csn5 (Protein ID: Pc_6891) with the counterpart of B. bassiana we have studied. However, a search through the NCBI protein database with the ID code (listed in their Table S1) led to no result. No matter whether the mentioned 11 chitosanases are CSN subunits or not, I found no reason to rename Csn5 in our study.

-I miss in the paper some implications of the present work for the practical usage of the fungus

Author response: The implication of our work has been has been shown by adding a sentence at the end of Conclusion.

-p.4l.192.  "larvae died post-CBI" should read"larvae dead post-CBI"

Author response: revised as suggested.

-p.5l.198-199. Why authors did not use insect cuticle as enzyme inducer for Bb?

Author response: With our long-term experience, bovine serum albumin is more efficient than smashed insect cuticle in inducing enzyme production in the submerged cultures.

-p.6l.16 I have always used disregulation but it seems that dysregulation is also correct.

Author response: Found in a huge website dictionary is dysregulation rather than disregulation.

-Fig.1E. Do authors have any explanation why ubi4 ubic3 ubc5 ubc 12 are up-regulated in deltacsn5?. Same for Fig1F in E#Ub-ligase genes6 and 15?

Author response: In B. bassiana, only ubi4 and ubr1 have been functionally characterized while ubi1/2 and ubi3 are essential for viability. Since most of those analyzed genes presumably functioning in the ubiquitin-proteasome pathway are unknown in function, it is hard to tell implications of those upregulated genes. However, we speculate that more E2 and two uspA-like genes repressed or abolished in the absence of csn5 could be causative of the increased ubiquitin accumulation in the mutant cells if one upregulated gene was counteracted by another downregulated significantly in the same family,

-Authors do not indicate (or I have not found it) specificity for PNA lectin

Author response: A sentence in the Results has been modified to indicate what carbohydate epitope was labeled by each lection.

-Fig. 2A why has the complemented mutant impaired pathogenicity?. The ECE abbreviation is not explained fully

Author response: Indeed, the WT and complemented mutant showed similar virulence via CBI despite a 0.2-day difference between their LT50 estimates. Thus, the virulence of the complemented mutant was largely (though not completely) restored in comparison to the virulence of the deletion mutant.

-Fig3G I would not use ultrastucture for SEM data (I would for TEM). Use SEM surface microstructure of conidia instead

Author response: Thanks! It has been changed to microstructure.

Reviewer 3 Report

In the present paper, the authors described the possible function of COP9 signalosome subunit 5 (Csn5) in insect pathogenicity and asexual development via ubiquitin-proteasome pathway. Manuscript is well-written and scientifically informative on basic molecular biological fields in insect pathogenic fungus Beauveria bassiana. I believe the manuscript is suitable publication in “Journal of Fungi” after revisions.

  1. Fig. 1C; The Western data is not acceptable especially in β-actin signals.
  2. Fig. 1E,F,G, Fig. 3D, Fig. 4B, E; Please indicate statistical significant.
  3. In the virulence assays (Fig. 2A), the authors did not perform any statistical analysis for these virulence data. Please provide proper information and data for statistical analysis in the virulence assays.
  4. What kind of SOD and catalase involved in oxidative stress response? To elevate of the level of paper, the authors need performed in gel assay for theses enzymes.

Author Response

Comments and Suggestions for Authors

In the present paper, the authors described the possible function of COP9 signalosome subunit 5 (Csn5) in insect pathogenicity and asexual development via ubiquitin-proteasome pathway. Manuscript is well-written and scientifically informative on basic molecular biological fields in insect pathogenic fungus Beauveria bassiana. I believe the manuscript is suitable publication in “Journal of Fungi” after revisions.

Author response: Thanks a lot for understanding and encouragement.

  1. Fig. 1C; The Western data is not acceptable especially in β-actin signals.

Author response: The β-actin signals are clear enough to measure signal intensity of each although not perfect. By the way, we have tried to slightly adjust the angle of the gel for better presentation.

  1. Fig. 1E,F,G, Fig. 3D, Fig. 4B, E; Please indicate statistical significant.

Author response: It is a trap to perform statistical analysis of transcript data because a 10% difference could be statistically significant but does not mean very much for the function of an analyzed gene. This is why transcriptomic analysis is usually performed at the significance of at least one-fold transcript change (50% down- or 100% up-regulation) to identify differentially expressed genes.

  1. In the virulence assays (Fig. 2A), the authors did not perform any statistical analysis for these virulence data. Please provide proper information and data for statistical analysis in the virulence assays.

Author response: Statistical analysis has been applied to the LT50 estimates as indices of fungal virulence via CBI but not applied to those via NCI, which was nearly abolished and hence resulted in no LT50 available for the deletion mutant (Fig. 2B).

  1. What kind of SOD and catalase involved in oxidative stress response? To elevate of the level of paper, the authors need performed in gel assay for theses enzymes.

Author response: The gel assays of B. bassiana SODs and catalases have been well shown in previous papers (Xie et al. 2012; Wang et al. 2012; Li et al. 2015), which have been cited wherever necessary in this manuscript.

Round 2

Reviewer 1 Report

Neither of my major comments were answered.

The authors did not change figure 1 and still claim that the UPS is involved  while this is not correct. The link between CSN5 and the UPS is highly studied. The question here should be whether CSN5 altered the pathogenicity of the fungi through the UPS. THIS HAS NOT BEED ANSWERED.  

  • Figure 1C – 1F show the effect of CSN5 on ubiquitination. This is not new and not related at all to the pathogenicity they show in figures 2 to 4. Accordingly – this figure is missleading and cannot stay in the main text since it confuses the reader. As I previously wrote (and even provided few doi numbers), CSN5 does not always exist in the complex, and has various effects on the ubiquitinome. In short, the information does not help us understand the story presented below, but is a beginning of a new story that is not told in this article.
  • The authors (still!!) do not provide the names of enzymes in figures F, G. Without adding the names, the figure is not publishable. Authors cannot hide information from readers. They could add table for example. In any case, the illustration cannot be published as is in a reputable journal as JoF.
  • Figure 1C-1F - this reviewer suggested to replace the provided data on  (which could be still located at the supplementary) with novel information on CSN5 and the CSN complex in this organism. It is the first time to publish a paper about the CSN in this organism, and the information is novel and required.
    1. How many subunits
    2. Alinment of CSN5 with CSN5s of other related organisms
    3. phylogenetic tree for CSN5
    4. etc...

If the authors want to emphasize the importance of ubiquitin, they should add information (as I suggested in the previous round of comments, and which I am sure they do not want to), in order to prove a link between Ub and the pathogenicity of the fungus, including alteration of this link in deltacsn5.

However, I will agree to accept the article for publication right after this section is corrected.

Author Response

The authors did not change figure 1 and still claim that the UPS is involved  while this is not correct. The link between CSN5 and the UPS is highly studied. The question here should be whether CSN5 altered the pathogenicity of the fungi through the UPS. THIS HAS NOT BEED ANSWERED.

Author response: Figure 1 has been modified by moving all western blot and transcript data to supplementary material (Figure S4) as suggested. However, we still think that it is meaningful to show an increased accumulation of intracellular free ubquitin in the absence of csn5. Although the mentioned link has been well studied, it is the first time to show it in a fungal insect pathogen. The revealed link helps to explain why Csn5 is indispensable for the fungal insect pathogenicity shown afterwards. Csn5 is not a true transcription factor that can be distinguished with sequence feature and is localized exclusively in nucleus. Only is the link revealed by the western blot able to explain a substantial role of Csn5 in transcriptional regulation of crucial genes and functional control of those proteins (shown with the activities of hydrophobins and cuticle-degrading enzymes in this study) collectively essential for the fungal insect pathogenicity. This is because a balance between ubiquitination and deubiquitination is a well-known mechanism underlying gene mediation and protein quality/function control.

  • Figure 1C – 1F show the effect of CSN5 on ubiquitination. This is not new and not related at all to the pathogenicity they show in figures 2 to 4. Accordingly – this figure is missleading and cannot stay in the main text since it confuses the reader. As I previously wrote (and even provided few doi numbers), CSN5 does not always exist in the complex, and has various effects on the ubiquitinome. In short, the information does not help us understand the story presented below, but is a beginning of a new story that is not told in this article.

Author response: Please see the above response.

  • The authors (still!!) do not provide the names of enzymes in figures F, G. Without adding the names, the figure is not publishable. Authors cannot hide information from readers. They could add table for example. In any case, the illustration cannot be published as is in a reputable journal as JoF.

Author response: The gene names have been added to the figure panels. The information of each analysed gene is listed in Table S2, and hence would not cause any confusion.

  • Figure 1C-1F - this reviewer suggested to replace the provided data on  (which could be still located at the supplementary) with novel information on CSN5 and the CSN complex in this organism. It is the first time to publish a paper about the CSN in this organism, and the information is novel and required.
    1. How many subunits
    2. Alinment of CSN5 with CSN5s of other related organisms
    3. phylogenetic tree for CSN5
    4. etc...

If the authors want to emphasize the importance of ubiquitin, they should add information (as I suggested in the previous round of comments, and which I am sure they do not want to), in order to prove a link between Ub and the pathogenicity of the fungus, including alteration of this link in deltacsn5.

Author response: Phylogenetic relationships of all CSN subunits between B. bassiana and two model fungi have been illustrated in added Figure S3.

However, I will agree to accept the article for publication right after this section is corrected.

Author response: Thanks a lot for your patience. We have tried our best to revise the manuscript as suggested.
